# Identification of Daphnane Diterpenoids from *Wikstroemia indica* Using Liquid Chromatography with Tandem Mass Spectrometry

**DOI:** 10.3390/plants12203620

**Published:** 2023-10-19

**Authors:** Mi Zhang, Kouharu Otsuki, Reo Takahashi, Takashi Kikuchi, Di Zhou, Ning Li, Wei Li

**Affiliations:** 1Faculty of Pharmaceutical Sciences, Toho University, Miyama 2-2-1, Funabashi 274-8510, Chiba, Japan; zhangmi495@gmail.com (M.Z.); takashi.kikuchi@phar.toho-u.ac.jp (T.K.); 2Key Laboratory of Innovative Traditional Chinese Medicine for Major Chronic Diseases of Liaoning Province, Key Laboratory for TCM Material Basis Study and Innovative Drug Development of Shenyang City, School of Traditional Chinese Materia Medica, Shenyang Pharmaceutical University, Shenyang 110016, China; zd930322@126.com (D.Z.); liningsypharm@163.com (N.L.)

**Keywords:** daphnane diterpenoids, LC-MS/MS, *Wikstroemia indica*, MS/MS fragmentation

## Abstract

Liquid chromatography coupled with tandem mass spectrometry (LC-MS/MS) has emerged as a powerful tool for the rapid identification of compounds within natural resources. Daphnane diterpenoids, a class of natural compounds predominantly found in plants belonging to the Thymelaeaceae and Euphorbiaceae families, have attracted much attention due to their remarkable anticancer and anti-HIV activities. In the present study, the presence of daphnane diterpenoids in *Wikstroemia indica*, a plant belonging to the Thymelaeaceae family, was investigated by LC-MS/MS analysis. As a result, 21 daphnane diterpenoids (**1**–**21**) in the stems of *W. indica* were detected. Among these, six major compounds (**12**, **15**, **17**, **18**, **20**, and **21**) were isolated and their structures were unequivocally identified through a comprehensive analysis of the MS and NMR data. For the minor compounds (**1**–**11**, **13**, **14**, **16**, and **19**), their structures were elucidated by in-depth MS/MS fragmentation analysis. This study represents the first disclosure of structurally diverse daphnane diterpenoids in *W. indica*, significantly contributing to our understanding of bioactive diterpenoids in plants within the Thymelaeaceae family.

## 1. Introduction

Liquid chromatography coupled with high-resolution tandem mass spectrometry (LC-HR-MS/MS), usually equipped with an electrospray ionization source, has high adaptability across a broad spectrum of compounds, offering high mass accuracy and sensitivity. Moreover, it provides information-rich fragmentation through product ion spectra, thereby potentially revealing details about the molecular formula and structure of diverse secondary metabolites found in plants [1]. The conventional phytochemical research process often necessitates substantial amounts of accessible plant materials and time-consuming purification procedures, whereas applying LC-MS/MS analysis on crude plant extracts at the early stage of phytochemical investigations allows for the rapid identification of the compounds [2,3].

Daphnane diterpenoids, characterized by their *trans*-fused 5/7/6-tricyclic skeleton, have garnered attention for their diverse biological activities, including anticancer [4], anti-HIV [5], analgesic [6], anti-inflammatory [7], and neurotrophic activities [8,9]. These diterpenoids are predominantly found in plants of the Thymelaeaceae and Euphorbiaceae families, with the majority of them sourced from the Thymelaeaceae family [10]. Previous phytochemical investigations on plants of the Thymelaeaceae family have reported the isolation of daphnane diterpenoids from 16 genera, such as *Daphne*, *Pimelea*, *Stellera*, and *Wikstroemia*. Among these, the *Wikstroemia* genus, comprising over 70 species, holds significant potential as a source of daphnane diterpenoids. Isolation of daphnane diterpenoids has hitherto been reported from a number of species, including *W. monticola* [11], *W. mekongenia* [12], *W. retusa* [13,14], *W. polyantha* [15], *W. chamaedaphne* [16,17,18], *W. chuii* [19], and *W. ligustrina* [20]. It is evident that the *Wikstroemia* genus remains relatively underexplored in the research into daphnane diterpenoids. 

*Wikstroemia indica* (L.) C. A. Mey. is a semi-evergreen shrub mainly distributed in southeastern China, which has long been used as a traditional Chinese medicine for the treatment of bronchitis, hepatitis, and cancer [21]. Recent studies have revealed that the extract of this plant exhibited antiallergic [22], anti-inflammatory [23], and antineoplastic properties [24], therefore heightening interest in its pharmacological exploration. While previous phytochemical investigations of *W. indica* have yielded coumarins [25], flavonoids [26], lignans [27], and sesquiterpenoids [28], the presence of daphnane diterpenoids has yet to be documented.

During our ongoing research aimed at discovering biological diterpenoids from plants of the Thymelaeaceae family [5,20,29,30], this study comprehensively examined and identified daphnane diterpenoids in the stems of *W. indica* using LC-MS/MS analysis.

## 2. Results and Discussion

### 2.1. Detection of Daphnane Diterpenoids in W. indica by LC-MS/MS

Due to the limited availability of plant material, the presence of daphnane diterpenoids in *W. indica* was initially examined by LC-MS/MS analysis. The criteria for validating that the detected peaks represented daphnane diterpenoids were established based on a synthesis of our previous studies and literature review [20,31,32]. These criteria included: (1) In the mass spectra, protonated molecular ions ([M + H]^+^) and/or ammonium adduct ions ([M + NH_4_]^+^) were observed in positive ion mode, while deprotonated molecular ions ([M–H]^−^) and/or formate adduct ions ([M + HCOO]^−^) were observed in negative ion mode. (2) In the product ion spectrum obtained from the protonated molecular ion as a precursor ion, a diagnostic ion at *m*/*z* 253 (C_17_H_17_O_2_) or 269 (C_17_H_17_O_3_) was observed in the positive ion mode [31]. (3) The characteristic C_17_ product ions derived from C_20_ skeletons with the neutral loss of C_3_H_4_O_2_ were observed [32]. (4) When the ion peaks originated from a macrocyclic daphnane orthoester (MDO), the second and third criteria were not applicable. Instead, product ion peaks derived from continuous losses of H_2_O and CO were observed at the mass range of *m*/*z* 250–350 and *m*/*z* 400–550, respectively [20].

To enhance the detecting sensitivity, a crude diterpenoid fraction was prepared from the 95% EtOH extract using a sequence of procedures, including EtOAc–H_2_O partition and Diaion HP-20 column chromatography. Subsequent LC-MS/MS analysis of the crude diterpenoid fraction, guided by the aforementioned criteria, resulted in the detection of three major daphnane diterpenoid peaks (**15**, **20**, and **21**), strongly suggesting the occurrence of daphnane diterpenoids in *W. indica* stems. It was noteworthy that detecting daphnane diterpenoids can be challenging due to their chromatographic behavior, which was sometimes similar to common plant constituents, such as fatty acids, acylglycerols, and chlorophyll [33]. 

To further enhance the sensitivity of LC-MS/MS detection of daphnane diterpenoids, a portion of the crude diterpenoid fraction underwent additional fractionation through gradient HPLC. As a result, a total of 21 daphnane diterpenoid peaks (**1**–**21**) were detected from three out of twelve subfractions (Appendix A). Importantly, all these peaks were subsequently confirmed to be present in the crude diterpenoid fraction through extracted ion chromatogram (XIC) analysis (Figure 1, Table 1).

### 2.2. LC-MS Guided Isolation and Structural Determination of Major Daphnane Diterpenoids

An LC-MS guided isolation was carried out to obtain daphnane diterpenoids. The crude diterpenoid fraction was subjected to ODS column chromatography and eluted with a stepwise gradient of MeOH–H_2_O. The fractions, in which daphnane diterpenoids were detected by LC-MS/MS analysis, were subjected to silica gel column chromatography and eluted with a gradient of *n*-hexane–EtOAc–MeOH–HCOOH. Those fractions containing daphnane diterpenoids were further purified by preparative HPLC and resulted in the isolation of six major daphnane diterpenoids (**12**, **15**, **17**, **18**, **20**, and **21**) (Figure 2).

The isolated compounds were identified by detailed NMR and MS spectroscopic analyses. In the ^1^H- and ^13^C-NMR spectra, the characteristic resonances for an isopropenyl moiety at *δ*_H_ 4.83–4.99 (H_a_-16), 4.94–5.17 (H_b_-16), and 1.72–1.84 (H_3_-17), an epoxy group at *δ*_H_ 3.32–3.57 (H-7), *δ*_C_ 60.0–61.0 (C-6), and 63.5–64.4 (C-7), and an orthoester group at 116.5–119.8 (C-1′), indicated that they were daphnane diterpenoids (Appendix A) [10]. 

For compounds **15**, **17**, and **21**, the characteristic resonances of *α*,*β*-unsaturated carbonyl moiety at *δ*_H_ 7.56–7.61 (H-1), *δ*_C_ 160.5–161.4 (C-1), 136.6–136.9 (C-2), and 209.5–209.9 (C-3) indicated they belong to orthoester daphnane diterpenoids (Appendix A). The presence of a decanoate moiety at C-1′ in **15** was deduced from the proton resonance for the aliphatic methylene multiplets at *δ*_H_ 1.23–1.93 and a terminal methyl triplet at *δ*_H_ 0.86 (H_3_-10′). The 2*E*,4*E*-tetradecadienylidyne moiety of **21** was defined from the proton resonances for a conjugated diene at *δ*_H_ 5.89 (d, *J* = 15.5 Hz, H-2′), 6.68 (dd, *J* = 15.5, 10.6 Hz, H-3′), 6.04 (dd, *J* = 15.2, 10.6 Hz, H-4′), and 5.83 (dt, *J* = 15.2, 7.2 Hz, H-5′), and a *n*-nonyl moiety including eight methylenes at *δ*_H_ 1.24–2.07 and a terminal methyl group at *δ*_H_ 0.86 (t, *J* = 6.9 Hz, H_3_-14′). The presence of cinnamoyloxy moiety of **17** was deduced from a *trans*-olefinic moiety at *δ*_H_ 6.35 (d, *J* = 15.9 Hz, H-2″) and 7.62 (d, *J* = 15.9 Hz, H-3″), and a phenyl moiety at *δ*_H_ 7.37–7.51 (each multiplet, H-5″ to H-9″), as well as the carbon resonance for an ester carbonyl at *δ*_C_ 165.8 (C-1″), and a 2*E*,4*E*-decadienylidyne moiety was confirmed by the proton resonances at *δ*_H_ 5.65 (d, *J* = 15.4 Hz, H-2′), 6.66 (dd, *J* = 15.4, 10.6 Hz, H-3′), 6.04 (dd, *J* = 15.1, 10.6 Hz, H-4′), and 5.85 (dt, *J* = 15.1, 7.0 Hz, H-5′), and a *n*-pentyl moiety including four methylenes at *δ*_H_ 2.08 (H-6′), 1.37 (H-7′), and 1.27 (H-8′, 9′), and a terminal methyl group at *δ*_H_ 0.87 (t, *J* = 6.8 Hz, H_3_-10′). The location of the cinnamoyl moiety at C-12 was confirmed by the HMBC correlation from H-12 to C-1″. Thus, **15**, **17**, and **21** were determined as simplexin (**15**) [41], 12-*O*-(*E*)-cinnamoyl-9,13,14-ortho-(2*E*,4*E*)-decadienylidyne-5*β*,12*β*-dihydroxyresiniferonol-6*α*,7*α*-oxide (**17**) [40], and huratoxin (**21**) [42].

On the other hand, compounds **12**, **18**, and **20** belong to MDOs since the resonances of H-1 were observed as the methine proton at *δ*_H_ 2.11–2.97 and the methyl proton resonances at *δ*_H_ 0.79–1.26 of H_3_-10′ were observed as doublet (Appendix A) [10]. The presence of benzoyl moieties of **12**, **18**, and **20** were determined by the aromatic proton resonances at *δ*_H_ 8.02–8.16 (H-2′,6′), 7.35–7.47 (H-3′,5′), and 7.53–7.59 (H-4′). The locations of benzoyl moieties at C-3, C-18, and C-7′ of **12**, C-20 of **18**, and C-3 of **20** were confirmed by the HMBC experiment. Thus, **12**, **18**, and **20** were determined as stelleralide G (**12**) [38], wikstromacrin (**18**) [20], and pimelea factor P_2_ (**20**) [41].

### 2.3. Identification of Minor Daphnane Diterpenoids by MS/MS Fragmentation Elucidation

To identify the minor daphnane diterpenoids (**1**–**11**, **13**, **14**, **16**, and **19**), which could not be isolated, MS/MS fragmentation elucidation was performed. These daphnane diterpenoids exhibited abundant ions in the product ion spectra derived from the protonated molecular ion as a precursor ion. Consequently, a detailed interpretation of the MS/MS fragmentation pathways in positive mode for these peaks was conducted (Figure 3 and Figure 4). The identification of those peaks was confirmed by the LC-MS data which were in full accordance with the corresponding compounds isolated in our previous studies (Table 1) [5,29,30].

In the product ion spectra of peaks **3** and **16**, the characteristic product ion was observed at *m*/*z* 253 (C_17_H_17_O_2_), which was produced by the loss of the 6,7-epoxy moiety, along with the oxymethylene at C-20, as a C_3_H_4_O_2_ unit due to cleavage occurring at the B-ring. This observation suggested that both **3** and **16** were daphnane diterpenoids lacking a substituent at C-12 (Figure 3A and Appendix A). Furthermore, the product ions at *m*/*z* 207 (C_14_H_23_O), 95 (C_6_H_7_O), and 81 (C_5_H_5_O) for peak **3**, and at *m*/*z* 179 (C_12_H_19_O), 95 (C_6_H_7_O), and 81 (C_5_H_5_O) for peak **16** indicated that a 2*E*,4*E*-tetradecadienoyl moiety was ester-linked to the C-ring in peak **3** and a 2*E*,4*E*-dodecadienoyl moiety was ester-linked in peak **16**. However, the molecular formula of peak **3** was 18 Da (H_2_O) larger than that of the orthoester daphnane, huratoxin (**21**) [42] and the fragment ion of [M + H–H_2_O]^+^ appeared with greater intensity in the mass spectrum of peak **3** (Appendix A). Based on these observations, it was concluded that peak **3** represented a polyhydroxy daphnane type compound, which lacks the orthoester moiety at the C-ring. Thus, peaks **3** and **16** were identified as wikstroelide M (**3**) [14] and wikstrotoxin B (**16**) [11], respectively.

In the product ion spectra of peaks **6**, **9**, **10**, **14**, and **19**, the product ion generated by the loss of the C_3_H_4_O_2_ unit was consistently observed at *m*/*z* 269 (C_17_H_17_O_3_), indicating that these peaks corresponded to orthoester daphnanes with a substituent attached to C-12 (Figure 3B and Appendix A). The product ions corresponding to substituents observed in these peaks were assignable as follows: a cinnamoyl moiety at *m*/*z* 131 (C_9_H_7_O), a coumaroyl moiety at *m*/*z* 147 (C_9_H_7_O_2_), a feruloyl moiety at *m*/*z* 177 (C_10_H_9_O_3_), a 2*E*,4*E*-decadienoyl moiety at *m*/*z* 151 (C_10_H_15_O), 95 (C_6_H_7_O), and 81 (C_5_H_5_O), a 2*E*,4*E*,6*E*-decatrienoyl moiety at *m*/*z* 149 (C_10_H_13_O) and 107 (C_9_H_9_O), and a 2*E*,4*E*-dodecadienoyl moiety at *m*/*z* 179 (C_12_H_19_O), 95 (C_6_H_7_O), and 81 (C_5_H_5_O). Namely, the feruloyl and 2*E*,4*E*,6*E*-decatrienoyl moieties were present in peak **6**, the coumaroyl and 2*E*,4*E*-decadienoyl moieties in peak **9**, the feruloyl and 2*E*,4*E*-decadienoyl moieties in peak **10**, and the cinnamoyl and 2*E*,4*E*,6*E*-decatrienoyl moieties in peak **14**. In peak **19**, only the product ions due to the 2*E*,4*E*-dodecadienoyl moiety were observed, but the molecular formula and the observation of product ions derived from the neutral loss of C_2_H_4_O_2_ suggested the presence of the acetyl moiety. Thus, peaks **6**, **9**, **10**, **14**, and **19** were identified as acutilobin C (**6**) [36], daphneodorin D (**9**) [29], acutilobin D (**10**) [36], 12-*O*-(*E*)-cinnamoyl-9,13,14-ortho- (2*E*,4*E,6E*)-decatrienylidyne-5*β*,12*β*-dihydroxyresiniferonol-6*α*,7*α*-oxide (**14**) [40], and wikstroelide A (**19**) [13].

Peaks **1**, **2**, **4**, **5**, **7**, **8**, **11**, and **13** were identified as MDOs by their characteristic MS/MS fragmentation patterns. Although the number of oxygen functional group varied among these compounds, they were all characterized by the abundance of C_30_ to C_28_ product ions observed in the range of *m*/*z* 400–550. The molecular formula of peak **7** indicated the absence of acyl groups. In the product ion spectrum, the neutral loss associated with the macrocyclic ring was assigned to be C_10_H_16_O as in compounds **18** and **20**. In addition, a series of C_20_ to C_18_ product ions were observed with successive losses of H_2_O and CO from *m*/*z* 327 (C_20_H_23_O_4_) (Figure 4A and Appendix A). Peak **7** was suggested to possess the cyclopentanone A-ring structure based on the degree of unsaturation and was identified as pimelea factor S_6_ (**7**) [37]. Peak **5** had a molecular formula that was 16 Da (OH) larger than peaks **18** and **20**. The product ion spectrum of peak **5** exhibited a series of C_20_ to C_18_ product ions below *m*/*z* 350, as observed in **18** and **20** (Figure 4B and Appendix A). However, the neutral loss associated with the macrocyclic ring differed from **18** and **20**, where it was C_10_H_16_O rather than C_10_H_14_O in **5**. These observations indicated that **5** possesses an additional hydroxyl group at C-2′ of the macrocyclic ring compared to **18** and **20**, and was further identified as kraussianin (**5**) [35].

The pair of peaks **1** and **2**, as well as the pair of peaks **8** and **11**, had the same molecular formula and exhibited similar product ion spectra, indicating that each pair, like compounds **18** and **20**, was in a regioisomeric relationship. The product ion spectra of peaks **1** and **2** revealed three molecules of C_7_H_6_O_2_ neutral loss originating from benzoyl acids and two molecules of C_2_H_4_O_2_ neutral loss originating from acetic acids, as well as the neutral loss of C_10_H_10_O associated with the macrocyclic ring. In addition, a series of C_30_ to C_28_ product ions were observed with successive losses of H_2_O and CO from *m*/*z* 507 (C_30_H_35_O_7_) (Figure 4C and Appendix A). These observations suggested that peaks **1** and **2** correspond to daphneodorin B (**1**) or daphneodorin C (**2**) with the same molecular formula and combinations of acyl groups. By comparison of retention times, peaks **1** and **2** were identified as daphneodorin C (**1**) and daphneodorin B (**2**), respectively [29]. The product ion spectra of peaks **8** and **11** revealed the elimination of two molecules of C_7_H_6_O_2_, indicating the presence of two benzoyloxy moieties (Figure 4C and Appendix A). Additionally, the neutral loss associated with the macrocyclic ring was assigned to be C_10_H_14_O as in **5** and the product ion pattern below *m*/*z* 350 was the same as **12**, which were identified as the regioisomer, stelleralide H (**8**) [38] and gnidimacrin (**11**) [39], respectively. 

Peak **4** had a molecular weight 14 Da greater than peak **7** but the neutral loss associated with the macrocyclic ring was assigned to be C_10_H_16_O as in peak **7**, suggesting that the daphnane skeleton was different from peak **7** (Figure 4D and Appendix A). Based on the molecular formula and degree of unsaturation, peak **4** was identified as pimelotide C [34], which has the bicyclo[2.2.1]heptane A-ring structure. In the product ion spectrum of peak **13**, the observation of a loss of C_7_H_6_O_2_ suggested the presence of a benzoyloxy moiety (Figure 4E and Appendix A). Furthermore, the product ions observed within the range of *m*/*z* 400–550 were 2 Da smaller than those of peak **4**. Thus, peak **13** was identified as stelleralide C [5], which shared the bicyclo[2.2.1]heptane A-ring structure of peak **4** and a benzoyloxy moiety attached to C-18.

## 3. Materials and Methods

### 3.1. General Experimental Procedures

The NMR spectra were collected on a JEOL ECA-500 spectrometer (JEOL Ltd., Tokyo, Japan) with the deuterated solvent used as the internal reference. The ^1^H-NMR spectra were performed at 500 MHz, and the ^13^C-NMR spectra were generated at 125 MHz. HRESIMS was conducted using a Q-Exactive Hybrid Quadrupole Orbitrap mass spectrometer (Thermo Scientific, Waltham, MA, USA). The following columns were utilized for column chromatography: Diaion HP-20 (Mitsubishi Chemical Corporation, Tokyo, Japan), ODS (Chromatorex DM1020T, Fuji Silysia Chemical Ltd., Aichi, Japan) and silica gel (Chromatorex PEI MB 100-40/75, Fuji Silysia Chemical Ltd., Aichi, Japan) columns. For gradient HPLC, two JASCO/PU-2080 Plus Intelligent HPLC pumps (JASCO Corporation, Tokyo, Japan), equipped with an MX-2080-32 dynamic mixer (JASCO Corporation, Tokyo, Japan), a JASCO UV-970 Intelligent UV/vis detector, and an SSC-6800 fraction collector (JASCO Corporation, Tokyo, Japan), were utilized. For preparative HPLC, a Waters 515 HPLC pump (Waters Corporation, Massachusetts, USA), equipped with an ERC RefractoMax520 differential refractometer detector (Thermo Scientific, Waltham, MA, USA) and a Shimadzu SPD-10A UV-vis detector (Shimadzu, Kyoto, Japan), was utilized. For normal-phase HPLC separations, a silica gel column (YMC-Pack SIL, 5 µm, 250 × 20 mm) was utilized with a flow rate of 5.0 mL/min. For reversed-phase HPLC separations, an RP-C_18_ silica gel column (YMC-Actus Triart C_18_, 5 µm, 150 × 20 mm) was utilized, with a flow rate of 8.0 mL/min.

### 3.2. Plant Material

The stems of *W. indica* were collected at Guangxi Province, People’s Republic of China in February 2018 and identified by Dong Liang (Kunming Plant Classification and Biotechnology Co., Ltd., Kunming, China). A voucher specimen (accession number: 20201021) had been deposited in the herbarium of Shenyang Pharmaceutical University.

### 3.3. Extraction and Isolation

The air-dried whole plants of *W. indica* (1000 g) were cut into small pieces and extracted with 95% EtOH at room temperature to give an EtOH extract and a residue. The EtOH extract was concentrated (63.0 g), suspended in H_2_O, and then partitioned with EtOAc. The EtOAc fraction (30.0 g) was subjected to Diaion HP-20 column chromatography, eluted with a stepwise gradient of MeOH/H_2_O (from 5:5 to 10:0, *v*/*v*) to afford three fractions (E1 to E3). The E3 fraction (10.8 g) was subjected to ODS column chromatography, eluted with a stepwise gradient of MeOH–H_2_O (from 7:3 to 10:0, *v*/*v*) to afford four subfractions (E3-1 to E3-4). Subfraction E3-2 (1840.3 mg) was subjected to silica gel column chromatography, eluted with a gradient of *n*-hexane–EtOAc–MeOH–HCOOH, to afford four subfractions (E3-2-1 to E3-2-5). Subfraction E3-2-3 (193.4 mg) was purified by RP-HPLC (70% CH_3_CN, 80% CH_3_CN) to give **12** (0.6 mg). Subfraction E3-2-2 (73.0 mg) was purified by RP-HPLC (85% CH_3_CN) to give six subfractions (E3-2-2-1 to E3-2-2-6). Subfraction E3-2-2-3 (16.6 mg) was purified by RP-HPLC (80% CH_3_CN), followed by NP-HPLC (*n*-hexane/AcOEt, 3:7) to give **15** (7.6 mg), **17** (1.6 mg), and **18** (1.3 mg). Subfraction E3-2-2-4 (4.4 mg) was purified by RP-HPLC (85% CH_3_CN) to give **20** (1.9 mg) and **21** (1.0 mg).

### 3.4. LC-MS/MS Conditions

The LC-MS/MS analysis was performed using the same instruments and column as in previous experiments [20]. For LC conditions, the mobile phase comprised eluent A (distilled water with 0.1% formic acid) and B (acetonitrile with 0.1% formic acid), programmed as follows: 0–15 min, a linear gradient from 50% to 100% B, 15–18 min, 100% B, followed by column re-equilibration at 50% B for 10 min before the subsequent injection. For MS conditions, the in-source CID was set at 0 eV, and the resolution was 70,000 for full MS and 35,000 for full MS/data dependent (dd)-MS/MS modes. The AGC was established at 1E6 for full MS and 1E5 for dd-MS/MS. Data-dependent scanning was performed using HCD with the normalized collision energy at 15 eV. The extracted ion spectra were generated by extracting the following base peaks of each compounds within ± 5 ppm mass tolerance: *m*/*z* 1028.4283 [M + NH_4_]^+^ (**1**), *m*/*z* 1028.4264 [M + NH_4_]^+^ (**2**), *m*/*z* 585 [M + H–H_2_O]^+^ (**3**), *m*/*z* 547.2888 [M + H]^+^ (**4**), *m*/*z* 655.3469 [M + H]^+^ (**5**), *m*/*z* 719.3062 [M + H]^+^ (**6**), *m*/*z* 533.3104 [M + H]^+^ (**7**), *m*/*z* 775.3690 [M + H]^+^ (**8**), *m*/*z* 691.3133 [M + H]^+^ (**9**), *m*/*z* 721.3203 [M + H]^+^ (**10**), *m*/*z* 775.3687 [M + H]^+^ (**11**), *m*/*z* 912.4147 [M + NH_4_]^+^ (**12**), *m*/*z* 667.3107 [M + H]^+^ (**13**), *m*/*z* 673.2999 [M + H]^+^ (**14**), *m*/*z* 533.3107 [M + H]^+^ (**15**), *m*/*z* 557.3109 [M + H]^+^ (**16**), *m*/*z* 675.3151 [M + H]^+^ (**17**), *m*/*z* 639.3527 [M + H]^+^ (**18**), *m*/*z* 643.3458 [M + H]^+^ (**19**), *m*/*z* 639.3525 [M + H]^+^ (**20**), and *m*/*z* 585.3409 [M + H]^+^ (**21**). All data collected in the profile mode were acquired and processed using Thermo Xcalibur 4.1 software.

## 4. Conclusions

This study represents the first comprehensive identification of 21 daphnane diterpenoids from the stems of *W. indica* through a combination of LC-MS guided isolation and MS/MS fragmentation elucidation. The investigation revealed that *W. indica* contained structurally diverse daphnane diterpenoids, including orthoester daphnane type, polyhydroxy daphnane type, and macrocyclic daphnane orthoester type compounds. The application of MS/MS fragmentation elucidation for structural analysis enabled the rapid and precise identification of these diterpenoids within crude plant extracts. This methodology holds great promise for future research endeavors aimed at discovering bioactive diterpenoids from plants of the Thymelaeaceae family.

## Figures and Tables

**Figure 1 plants-12-03620-f001:**
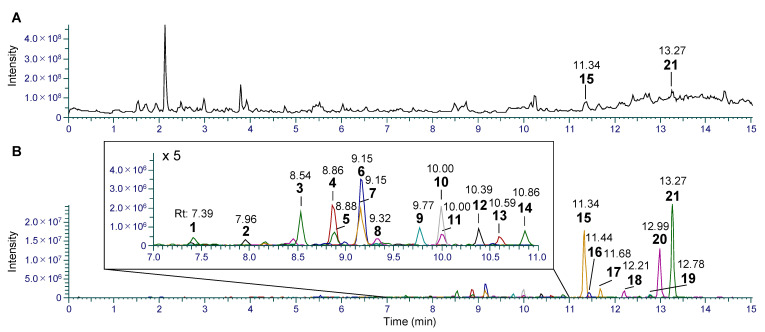
(**A**) Total ion chromatogram in the positive ion mode and (**B**) extracted ion chromatogram from the crude diterpenoid fraction of the stems of *W. indica*.

**Figure 2 plants-12-03620-f002:**
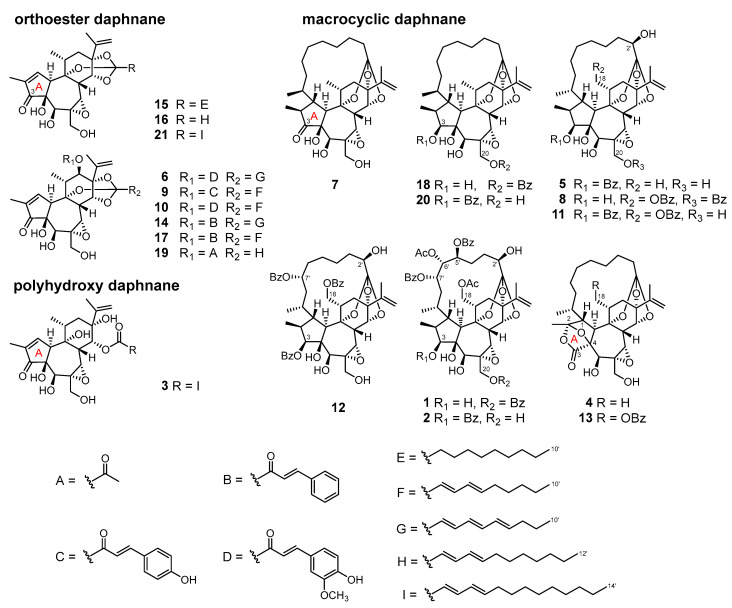
Structures of daphnane diterpenoids **1**–**21**.

**Figure 3 plants-12-03620-f003:**
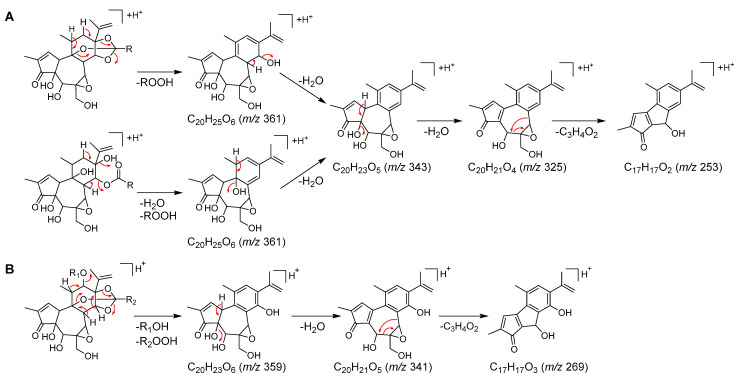
Proposed ESI-MS/MS fragmentation pathways for minor daphnane diterpenoids in positive mode. (**A**) peaks **3** and **16**, and (**B**) peaks **6**, **9**, **10**, **14**, and **19**.

**Figure 4 plants-12-03620-f004:**
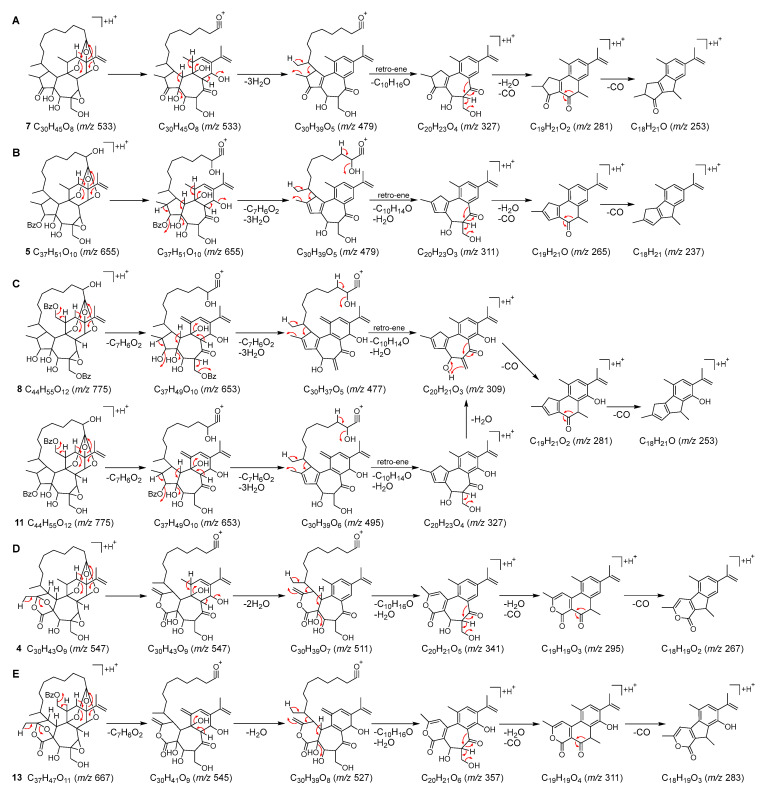
Proposed ESI-MS/MS fragmentation pathways for minor daphnane diterpenoids in positive mode. (**A**) peak **7**, (**B**) peak **5**, (**C**) peaks **8**, **11**, (**D**) peak **4**, and (**E**) peak **13**.

**Table 1 plants-12-03620-t001:** Daphnane diterpenoids **1**–**21** identified from the stems of *W. indica*.

No.	Rt (min)	Molecular Formula	[M + H]^+^ (*m*/*z*)		ESI-MS/MS (*m*/*z*) (%) ^c^	Identification
			Detected Mass (*m*/*z*)	Error (ppm)		
**1** ^a^	7.39	C_55_H_62_O_18_	1011.3984	−2.97	793 (36), 775 (22), 765 (19), 747 (20), 731 (34), 703 (23), 689 (16), 671 (47), 659 (12), 653 (17), 643 (17), 625 (24), 609 (18), 567 (24), 549 (100), 531 (15), 521 (39), 507 (36), 503 (37), 493 (10), 489 (66), 479 (18), 477 (18), 471 (37), 461 (68), 459 (23), 443 (80), 433 (24), 431 (25), 425 (23), 415 (31), 403 (27), 397 (19), 375 (22), 363 (18), 339 (23), 325 (16), 307 (37), 295 (22), 291 (21), 279 (37), 263 (22), 221 (21), 183 (18), 181 (47), 153 (16), 141 (25), 105 (65)	daphneodorin C [29]
**2** ^a^	7.96	C_55_H_62_O_18_	1011.3984	−2.43	793 (52), 775 (30), 765 (19), 747 (19), 731 (22), 707 (16), 689 (19), 671 (57), 653 (23), 643 (20), 629 (21), 625 (23), 611 (28), 593 (18), 583 (17), 549 (72), 531 (17), 521 (22), 507 (29), 503 (20), 489 (73), 479 (22), 471 (54), 461 (68), 453 (22), 443 (56), 425 (18), 415 (19), 375 (15), 363 (29), 307 (37), 291 (26), 279 (16), 221 (25), 181 (47), 163 (41), 141 (22), 105 (100)	daphneodorin B [29]
**3** ^a^	8.54	C_34_H_50_O_9_	603.3527	0.36	585 (29), 361 (10), 343 (23), 325 (20), 315 (6), 313 (5), 307 (10), 297 (16), 295 (5), 279 (11), 271 (7), 269 (6), 267 (8), 253 (15), 207 (100), 203 (5), 107 (8), 95 (8), 81 (6)	wikstroelide M [14]
**4** ^a^	8.86	C_30_H_42_O_9_	547.2888	−2.45	529 (35), 511 (30), 501 (32), 493 (27), 483 (59), 467 (29), 465 (64), 449 (21), 447 (29), 439 (25), 437 (40), 423 (23), 421 (25), 419 (29), 405 (28), 395 (20), 341 (27), 323 (24), 295 (26), 283 (29), 255 (29), 239 (21), 236 (23), 235 (100), 233 (47), 227 (22), 215 (36), 203 (28), 199 (21), 193 (24), 187 (27), 161 (29), 135 (36), 133 (33)	pimelotide C [34]
**5**	8.88	C_37_H_50_O_10_	655.3469	−1.24	619 (10), 515 (28), 497 (79), 479 (92), 469 (38), 467 (14), 461 (21), 451 (100), 443 (16), 439 (11), 433 (71), 423 (24), 421 (12), 415 (20), 405 (27), 367 (12), 311 (19), 293 (45), 275 (16), 265 (31), 263 (10), 251 (14), 247 (10), 225 (10), 211 (16), 133 (18), 123 (11), 105 (77)	kraussianin [35]
**6** ^a^	9.15	C_40_H_46_O_12_	719.3062	−1.33	507 (3), 489 (4), 359 (4), 341 (12), 323 (18), 311 (3), 305 (4), 295 (12), 277 (4), 269 (8), 177 (75), 149 (95), 121 (11), 107 (100), 81(3)	acutilobin C [36]
**7** ^a^	9.15	C_30_H_44_O_8_	533.3104	−1.00	515 (36), 497 (66), 479 (100), 469 (28), 467 (16), 461 (66), 451 (53), 449 (25), 443 (20), 433 (59),425 (24), 423 (13), 421 (16), 415(21), 407 (14), 405 (21), 403 (16), 309 (14), 291 (12), 281 (13), 263 (12), 211 (13), 187 (19), 185 (16), 159 (13), 135 (13), 133 (27)	pimelea factor S_6_ [37]
**8** ^a^	9.32	C_44_H_54_O_12_	775.3690	0.20	635 (15), 617 (21), 599 (22), 545 (63), 527 (33), 495 (47), 477 (85), 459 (100), 449 (21), 447 (28), 441 (37), 431 (50), 429 (20), 423 (90), 419 (18), 413 (29), 405 (24), 401 (23), 319 (18), 309 (24), 291 (24), 281 (18), 279 (23), 263 (19), 251 (18), 151 (27), 105 (88)	stelleralide H [38]
**9** ^a^	9.77	C_39_H_46_O_11_	691.3133	−2.63	509 (4), 505 (2), 491 (5), 359 (7), 341 (20), 323 (21), 313 (4), 311 (3), 305 (3), 297 (2), 295 (13), 277 (4), 269 (9), 267 (3), 261 (3), 151 (100), 147 (75), 133 (3)	daphneodorin D [29]
**10** ^a^	10.00	C_40_H_48_O_12_	721.3203	−1.60	509 (4), 491 (7), 359 (6), 341 (22), 323 (23), 313 (3), 311 (3), 305 (5), 295 (16), 277 (4), 269 (11), 267 (4), 261 (3), 177 (95), 151 (100), 95 (6), 81 (6)	acutilobin D [36]
**11** ^a^	10.00	C_44_H_54_O_12_	775.3687	−0.09	563 (28), 545 (100), 513 (35), 495 (60), 477 (41), 467 (45), 465 (24), 449 (78), 441 (16), 437 (35), 431 (34), 425 (20), 423 (20), 421 (24), 419 (19), 391 (18), 309 (17), 291 (19), 263 (19), 105 (40)	gnidimacrin [39]
**12** ^b^	10.39	C_51_H_58_O_14_	912.4147	−1.98	773 (48), 651 (47), 633 (58), 615 (34), 543 (100), 511 (57), 493 (82), 481 (20), 475 (58), 465 (54), 463 (36), 447 (90), 435 (43), 429 (38), 421 (47), 419 (21), 417 (22), 327 (23), 309 (40), 291 (26), 279 (22), 105 (55)	stelleralide G [38]
**13** ^a^	10.59	C_37_H_46_O_11_	667.3107	−0.93	545 (100), 527 (66), 509 (59), 499 (24), 491 (37), 483 (26), 481 (52), 465 (55), 463 (37), 453 (23), 447 (28), 445 (31), 435 (35), 419 (21), 417 (21), 357 (23), 321 (27), 295 (22), 235 (96), 231 (61), 203 (54), 185 (27), 173 (20), 153 (21), 105 (45)	stelleralide C [5]
**14** ^a^	10.86	C_39_H_44_O_10_	673.2999	−1.27	359 (1), 341 (3), 323 (4), 295 (3), 277 (1), 269 (2), 149 (100), 131 (6), 107 (21)	12-*O*-(*E*)-cinnamoyl-9,13,14-ortho-(2*E*,4*E*,6*E*)-decatrienylidyne-5*β*,12*β*-dihydroxyresiniferonol-6*α*,7*α*-oxide [40]
**15** ^b^	11.34	C_30_H_44_O_8_	533.3107	−0.23	361 (9), 343 (24), 325 (49), 307 (35), 297 (28), 279 (30), 267 (58), 253 (100), 203 (16), 155 (6)	simplexin [41]
**16**	11.44	C_32_H_44_O_8_	557.3109	−0.08	361 (8), 343 (23), 325 (33), 313 (7), 307 (30), 297 (19), 295 (11), 285 (6), 279 (20), 277 (6), 267 (55), 261 (6), 253 (77), 251 (7), 249 (11), 225 (6), 179 (100)	wikstrotoxin B [11]
**17** ^b^	11.68	C_39_H_46_O_10_	675.3151	−1.85	675 (9), 509 (2), 507 (2), 491 (2), 359 (5), 341 (15), 323 (21), 313 (4), 311 (3), 305 (4), 295 (13), 277 (5), 269 (11), 267 (3), 265 (2), 261 (2), 249 (2), 241 (2), 237 (2), 209 (2), 151 (100), 133 (4)	12-*O*-(*E*)-cinnamoyl-9,13,14-ortho-(2*E*,4*E*)-decadienylidyne-5*β*,12*β*-dihydroxyresiniferonol-6*α*,7*α*-oxide [40]
**18** ^a^	12.21	C_37_H_50_O_9_	639.3527	−0.10	621 (12), 499 (30), 481 (93), 463 (100), 453 (32), 445 (35), 436 (23), 435 (57), 417 (34), 407 (15), 311 (5), 293 (16), 265 (23), 105 (67)	wikstromacrin [20]
**19** ^a^	12.78	C_36_H_50_O_10_	643.3458	−2.97	365 (2), 359 (4), 341 (11), 323 (11), 313 (3), 305 (2), 295 (10), 277 (4), 269 (6), 267 (3), 207 (100), 189 (3), 107 (3)	wikstroelide A [13]
**20** ^b^	12.99	C_37_H_50_O_9_	639.3525	−0.28	621 (16), 499 (28), 481 (73), 463 (100), 453 (31), 451 (12), 445 (55), 435 (55), 433 (14), 423 (12), 417 (34), 409 (12), 407 (10), 405 (16), 311 (13), 293 (24), 275 (12), 265 (24), 251 (12), 133 (12), 105 (99)	pimelea factor P_2_ [41]
**21** ^b^	13.27	C_34_H_48_O_8_	585.3409	−2.14	361 (14), 343 (20), 325 (41), 313 (10), 307 (25), 297 (23), 295 (13), 279 (25), 267 (59), 253 (99), 249 (16), 207 (100)	huratoxin [42]

^a^ Identifications were confirmed with the daphnane diterpenoids isolated in our previous studies. ^b^ Isolated daphnane diterpenoids in this study. ^c^ ESI-MS/MS of [M + H]^+^ ion for peaks **1**–**11** and **13**–**21** and ESI-MS/MS of [M + NH_4_]^+^ ion for peak **12**.

## Data Availability

All new research data were presented in this contribution.

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
