# Peer review of "Identification of Daphnane Diterpenoids from Wikstroemia indica Using Liquid Chromatography with Tandem Mass Spectrometry"

_plants, 2023, doi:10.3390/plants12203620_

Round 1
Reviewer 1 Report
The authors used NMR and LC-MS/MS to study the Daphnane Diterpenoids from Wikstroemia indica. The NMR data together with LC-MS/MS achieved the structural elucidation of several Daphnane Diterpenoids. For those with very low abundance, tandem mass spectrometric experiments were used to propose the possible structures of the species. The experiment is well designed, and the data were carefully analyzed. The manuscript is well-written, and the data provided is comprehensive. Only a few suggestions here:
1. In section 2.1, criteria 2 still needs to be further explained here in this manuscript to make the manuscript easy to understand, even though references were provided.
2. In the methods section, the authors need to provide the details for the mass spectrometric experiments, for example, the fragmentation method used (CID/HCD?) key experimental parameters, resolutions, etc.
Reviewer 2 Report
Identification of Daphnane Diterpenoids from Wikstroemia indica by Liquid Chromatography with Tandem Mass Spectrometry
Wei Li et al reported 21 daphnane diterpenoids (1–21) in the stems of W. indica. Among these, six major compounds (12, 15, 17, 18, 20, and 21) were isolated and their structures were unequivocally identified through a comprehensive analysis of the MS and NMR data. For the minor compounds (1–11, 13, 14, 16, and 19), their structures were elucidated by in-depth MS/MS fragmentation analysis. This study represents the first disclosure of structurally diverse daphnane diterpenoids in W. indica, significantly contributing to our understanding of bioactive diterpenoids in plants within the Thymelaeaceae family.
The content of this study presents a novel and promising approach for the rapid evaluation and identification of components from the stems of W. indica. I recommend some minor revisions for this manuscript. My comments are as follows:
1. Please provide information on the pharmacology of the extract from W. indica, as the introduction of W. indica lacks sufficient detail.
2. Please provide clearer LC-MS/MS conditions, including detailed information about the detector, such as whether it is UV or another type.
3. Please provide a detailed program for NP-HPLC to separate the compounds.
Minor editing of English language required
Reviewer 3 Report
In this manuscript, the authors presented the results of their research on Identification of Daphnane Diterpenoids from Wikstroemia indica by Liquid Chromatography with Tandem Mass Spectrometry.
The manuscript is well written and all the results are presented clearly. However, I would like to see information about the purity of the selected fractions and their Physical and spectroscopic data according to the ACS's Preparation and Submission of Manuscripts standard.
